# Coaching doctors to improve ethical decision-making in adult hospitalized patients potentially receiving excessive treatment: Process evaluation study of the CODE intervention in doctors and nurses working in ten acute hospital wards

Ruth Piers[1,2‡]*, Let Dillen[2‡], Katrijn Goethals[3], An Lievrouw[4], Karen Versluys[1], Aglaja De Pauw[5], Celine Jacobs[5], Ine Moors[6], Fritz Offner[6], Anja Velghe[1,2], Nele Van Den Noortgate[1,2], Pieter Depuydt[7], Patrick Druwé[7], Dimitri Hemelsoet[8], Alfred Meurs[8], Jiska Malotaux[9], Wim Van Biesen[10], Francis Verbeke[10], Eric Derom[11], Dieter Stevens[11], Michel De Pauw[12], Fiona Tromp[12], Hans Van Vlierberghe[13], Karen Geboes[4,13], Frank Manesse[14,15], Stijn Vanheule[16], Dominique D. Benoit[7]

**1** Department of Geriatrics, Ghent University Hospital, Belgium, **2** Palliative Care Unit, Ghent University Hospital, Belgium, **3** Human Resources, Ghent University Hospital, Belgium, **4** Cancer Center, Ghent University Hospital, Belgium, **5** Department of Medical Oncology, Ghent University Hospital, Belgium, **6** Department of Hematology, Ghent University Hospital, Belgium, **7** Department of Intensive Care Medicine, Medical Unit, Ghent University Hospital, Belgium, **8** Department of Neurology, Ghent University Hospital, Belgium, **9** Department of General Internal Medicine and Infectious Diseases, Ghent University Hospital, Belgium, **10** Department of Nephrology, Ghent University Hospital, Belgium, **11** Department of Respiratory Medicine, Ghent University Hospital, Belgium, **12** Department of Cardiology, Ghent University Hospital, Belgium, **13** Department of Gastro-enterology and Hepatology, Ghent University Hospital, Belgium, **14** Independent, Conversio, Ghent, Belgium, **15** Kets de Vries Institute, London, United Kingdom, **16** Faculty of Psychology and Educational Sciences, Ghent University, Ghent, Belgium

‡ Shared first authorship.
* ruth.piers@uzgent.be

## Abstract

### Introduction

High quality hospital care for serious ill persons requires exemplary team-based communication and decision-making. However, there is a need for evidence-based interventions in this area. This study's aim is to gain insight in the adoption and implementation of the COaching Doctors to improve Ethical decision-making (CODE) intervention, composed of one-on-one professional coaching in doctors and an alert to identify patients potentially receiving excessive treatment.

### Methods

Mixed-method process evaluation study using focus group interviews, sub-analysis of the ethical decision-making climate questionnaire (EDMCQ) in nurses and doctors and a satisfaction survey in doctors.

**Data availability statement:** Due to ethical restrictions (in the informed consent of participants, it was stated that interview data will not be shared due to the potentially identifying and sensitive nature of the data). Data requests may be sent to ethisch.comite@uzgent.be (Ethics Committee of the Ghent University Hospital BC-09828).

**Funding:** This study is supported by grants from the Ghent University Hospital "Fonds voor Innovatie en Wetenschappelijk Onderzoek" and the Belgian "Fonds voor Wetenschappelijk Onderzoek" FWO (senior clinical investigators grant 1800518N obtained by Benoit in 2017). The funders had no role in study design, data collection and analysis, decision to publish, or preparation of the manuscript.

## Results

Of the 423 clinicians, 198 (60.2%) nurses and 63 (67.0%) doctors filled-out the EDMCQ and 20 nurses and 12 doctors participated to the focus-group interviews respectively. Thirty-five (70%) of the 50 coached doctors filled-out the satisfaction survey.

Doctors estimated their leadership skills higher than the nurses prior to the study period (P = 0.006) but no longer after the study period (P = 0.366). Nurses estimated ethical awareness and reflection higher (respectively, P = 0.002 and P = 0.049) and non-avoiding end-of-life decisions and active involvement of nurses lower than the doctors (respectively P < 0.001 and P = 0.023) after the study period but not before the study period (all P-values >0.05).

Doctors testified of increased awareness in ethical decision-making, more open interactions with patients and nurses, and more courage to intervene instead of doing nothing. Nurses spoke of an increased awareness to act as patient advocate, increased deliberation among peer nurses and frustration when they felt no effect of activating the CODE alert. Both doctors and nurses advised for more active involvement of nurses.

The mean overall satisfaction score in doctors was 8.69 on 10.

## Conclusion

The CODE study intervention was effective in raising awareness and self-reflection, yet did not succeed to close the (communication) gap between doctors and nurses in team-based ethical decision-making.

## Trial registration

ClinicalTrials.gov identifier: NCT 05167019.

## Introduction

High quality hospital care for serious ill persons requires exemplary team-based communication and decision-making [1,2]. Doctors have a pivotal role in fostering a safe ethical decision-making climate [3–8] crucial for optimizing end-of-life care management [1,4,9,10]. The COaching Doctors to improve Ethical decision-making (CODE) intervention aimed to improve team-based ethical decision-making in hospital wards in order to reduce potentially excessive treatments causing suffering in patients [1,11–13] and moral distress in clinicians taking care for patients at the end of life [9,14,15]. Coaching was chosen as a way for personal and professional growth of doctors in self-reflective and empowering leadership [16–18]. This method is firmly established in the sports and business world and is taking its first steps into the medical world [18,19].

Benoit et al. reported that the CODE intervention increased doctors' goal-oriented care via written do-not-intubate and do-not attempt cardiopulmonary resuscitation (DNI-DNACPR) decisions without concomitant increase in the quality of the ethical climate [17] as measured via de validated ethical decision-making climate questionnaire (EDMCQ) [4,5]. However, we lack insight in the underlying reasons for the observed outcomes: how did the CODE intervention work (or not) and what were the underlying mechanisms? Moreover, as the CODE study is the first study assessing the impact of coaching doctors in ethical decision-making, this is an unique opportunity to better understand how the intervention, and more specifically professional coaching, was perceived and implemented by nurses and doctors working in different wards of a large university hospital.

The research aim is to gain insight in the adoption, implementation and the intention of the participants to maintain the CODE intervention in future practice [16,20–22].

## Materials and methods

Process evaluation study based on the RE-AIM framework [22] of the stepped-wedge cluster randomized controlled CODE trial, conducted in the medical ICU and 9 referring internal medicine wards of Ghent University Hospital between February 2022 and February 2023.

The CODE intervention was composed of three main intervention components: (1) one-on-one coaching of doctors in self-reflective and empowering leadership as well as in team dynamics with regard to ethical decision-making, (2) the CODE alert to allow identification of patients potentially receiving excessive treatment by bedside clinicians (both nurses and doctors), and (3) observation of and feedback on interdisciplinary meetings. Nurses and doctors were instructed to use the CODE alert in the electronic patient file for each patient under their care in whom they deemed that treatment was excessive (defined as to be no longer consistent with the expected survival or quality of life or that is provided against the patient's or relatives' wishes). When two or more clinicians (nurses and/or doctors) perceived excessive treatment in a patient, the coach and the treating doctor were informed by an e-mail throughout the 4-month intervention period. Detailed description of the CODE intervention and the reach and effectivity of the study can be found elsewhere [16,17].

### Design and data collection

Recruitment of nurses and doctors for this process evaluation study started 01/01/2022 and ended 07/10/2023 (last interview for the qualitative part).

For the satisfaction survey, all 50 doctors who took part in the coaching received an e-mail with a link to a REDCap [23] survey hosted at the Ghent University Hospital after the 4-months intervention period in their respective ward. The survey consisted of a numeric rating scale from 0 to 10 for overall satisfaction, questions pertaining the organization and used coaching techniques as described in the coaching protocol [16], as well as questions pertaining gained awareness and skills in ethical decision-making based on theoretical frameworks about conversations in seriously ill patients [8,24–26].

All nurses and doctors were invited to fill out the EDMCQ [4,5] before and after the 12 months intervention period (February 2022 until February 2023). We refer to earlier publications for more details on the EDMCQ tool and data collection method [16,17].

In addition, a qualitative study design was set up to get insight into the experiences of nurses and doctors. For the interview study, the 261 clinicians who filled-out the EDMCQ were invited to participate by e-mail. Two reminders were sent to obtain a minimum of participants of each ward and of each profession. Data collection was driven by an interview guide that was composed by the head research investigators (RP, DB), and an independent interviewer (KV, PhD candidate, experienced geriatric nurse specialist, female). The interview guide was built thematically as the aim was to stir up group discussion and free speech. The focus was on enabling participants to share their lived experiences regarding the CODE intervention in a safe manner. Active listening, paraphrasing and open questioning made up the interviewer's stance. The interviewer proposed herself to the participants as a nurse with special interest in the topic of ethical decision-making but

independent from the CODE study. The focus groups were audio-recorded with written informed consent of the participants. The focus groups were transcribed verbatim and analyzed iteratively. The institutional review board requires that audio-recordings are destroyed after transcription. Analysis was performed before the primary outcomes of the CODE study were known. Analysis was led by the research question, viz. the experiences with the three aforementioned CODE intervention elements. Transcripts were read an experienced qualitative researcher who was independent from the CODE intervention (LD, PhD, psychologist, female) and by RP (MD, PhD, experienced geriatrician, female). Thematic analysis was applied using an open inductive coding framework inspired by the QUalitative Analysis Guide Of Leuven [27]. No software was used to manage the data (paper coding by LD). Data collection happened in two waves. After the first wave a preliminary analysis was discussed with the research team (KV: interviewer, AL: PhD, psychologist, CODE coach, female,; KG: certified coach, CODE coach, female) ensuing in researcher's triangulation. Adaptations were made to the interview guide based on themes that remained unsaturated. After the second wave a new round of analysis was performed, confronting the new data with the first analysis. This was again debated in group (RP, AL, KG, FM: senior certified coach, CODE supervisor, male; DB: MD, PhD, experienced intensivist, male). Finally, the other authors (some were doctors on the participating wards for participant checking) critically revised the text of analysis. Reporting is based on COREQ guidelines (S1Appendix) [28].

Finally, a sub-analysis of the validated EDMCQ across roles was performed.

### Ethics

This study has been reviewed and approved by the Ethics Committee of the Ghent University Hospital (BC-09828, date of approval 27/05/2021). A written informed consent was required for clinicians for the interview study and an electronic informed consent for the satisfaction survey and the EDMCQ.

## Results

### Participants

Thirty-five of 50 doctors (response rate 70%) who participated in one-on-one coaching filled out the satisfaction survey. For the interview study, we attained 32 participants (20 nurses and 12 doctors). More characteristics can be found in Table 1 and S1 Table.

In total 270 and 261 of 423 clinicians filled-in the EDMCQ before and after the study period (response rate 67.0% in doctors and 60.2% in nurses). Characteristics of the participants can be found in an earlier publication [17].

### Satisfaction survey

Participating doctors gave a mean satisfaction score of 8.69 (SD, 0.796). Over 80% indicated the coaching made them more aware of how they deal with conflicts and difficult decision-making, they gained insight in the perspectives of team-members, and stated to use the learned skills in daily practice (Table 2). The items that improved to a lesser extent were collaboration among doctors: collaboration with senior (60%), junior (60%) and referring doctors (51%) (Table 2). Ninety-seven percent found the one-on-one coaching useful for themselves; 60% said it increased their job satisfaction (Table 2).

### Interview study

In the first and second wave of qualitative research, respectively five and three focus groups were held. Additionally, two individual interviews were organized because the participants could not attend the focus groups (S1 Table). The focus groups lasted on average 59 minutes. Data saturation was reached after these two waves.

The results are structured around the diverse elements of the intervention: (A) one-on-one coaching, (B) CODE alert and (C) observation of interdisciplinary team meetings.

**Table 1. Characteristics describing the attained samples.**

| Characteristic | Satisfaction survey (*N*=35) Frequency *n* (%) | Qualitative study (*N*=32) Frequency *n* (%) |
|---|---|---|
| Gender | | |
| Male | 17 (49%) | 8 (26%) |
| Female | 18 (51%) | 23 (74%) |
| Age | | |
| 20–29 years | 9 (26%) | 3 (9%) |
| 30–39 years | 13 (37%) | 10 (31%) |
| 40–49 years | 8 (23%) | 12 (38%) |
| >50 years | 5 (14%) | 7 (22%) |
| Discipline | | |
| Junior doctor | 13 (37%) | 3 (9%) |
| Senior doctor | 22 (63%) | 9 (28%) |
| Nurse | NA | 16 (50%) |
| Deputy head nurse | NA | 2 (6%) |
| Head nurse | NA | 2 (6%) |
| Wave of interviews | | |
| Wave 1 | NA | 19 (59%) |
| Wave 2 | NA | 13 (41%) |

Legend: NA=non applicable; 1 missing value for gender in qualitative study.

**One-on-one coaching.** *The coaching process:* Participating doctors used the coaching sessions for debriefing challenging patient cases, team conflicts and for reflection on patient cases in which the CODE alert was activated (quotations [Q]1, Table 3). Participants testified of a **process** wherein their **perspective was broadened** (Q2-3; Table 3). Although reflection is inherent to a professional stance and thus not entirely new to most participants, the reflection process within the coaching sessions was from a whole different nature. It was described as more philosophical and personal. As most participants were not familiar with such a **personal approach** in their work context, this asked for a subtle process in which the coach build trust and confidentiality. This process was time-intensive and often evolved from an initially defensive posture, to **step by step allowing vulnerability** (Q4-5, Table 3), leaving the habitual stance of defense behind (Q6, Table 3).

*Perceived impact of the coaching on doctors:* Participants characterized the coaching sessions as enlightening and transformative. Some doctors expressed professional and personal growth. It opened up possibilities to look in a meta-level manner at how doctors deal with (in)**vulnerability** (Q7, Table 3). Effects were articulated on several domains.

*Expansion of ways to interact:* Many participants testified acquiring a more open and explorative attitude in interaction (Q8-9, Table 3). Some participants said they learned how to communicate in a more assertive way and to name things in a non-attacking way (Q10-11, Table 3).

*Increase of awareness:* Participants exhibited increased sensitivity to team dynamics and their own contributions therein, as well as a greater awareness of their blind spots, routines and behavior patterns (Q12, Table 3).

*Increase of one's leadership:* Participating doctors mentioned that they were more prone to take up their leadership responsibility by daring to explicitly intervene instead of doing nothing (Q13, Table 3).

Although most participants testified that the coaching had impact in some way, the sustainability of the effect was nuanced (Q14, Table 3). Some doctors were uncertain about the visibility of the coaching's effects within their teams. However, in the focus groups with nurses it became clear that the effect of the coaching was felt on some wards (Q15-16,

**Table 2.  Satisfaction of participating doctors to the one-on-one coaching (N = 35).**

| | Frequency *n (%)* |
|---|---|
| **Organizational aspects** | |
| Total number of coaching sessions | |
| Appropriate | 23 (66%) |
| Not enough | 10 (28%) |
| Too much | 2 (6%) |
| Duration of one-on-one coaching sessions (appropriate) | 35 (100%) |
| **Coaching techniques** | |
| The coach enhanced self-reflection by promoting moments of silence (agree) | 35 (100%) |
| The coach asked questions that stimulate awareness (agree) | 35 (100%) |
| The coach ask questions to clarify the context (agree) | 35 (100%) |
| The coach summarizes the conversation on a regularly base (agree) | 33 (94%) |
| The coach give feedback about my approach (agree) | 34 (97%) |
| The coach paraphrases and proposes hypotheses (agree) | 34 (97%) |
| The coach provides tips and tricks to improve communication (agree) | 34 (97%) |
| The coach shares own experiences (agree) | 18 (51%) |
| The coach gives specific advice (agree) | 32 (91%) |
| The coach suggests concrete actions and objectives (agree) | 32 (91%) |
| The coach tells me what I should not do (agree) | 26 (74%) |
| **Effects of the coaching with regard to ethical decision-making: awareness** | |
| I have more insight in how I cope with prognostic uncertainty (agree) | 22 (63%) |
| I have more insight in how I cope with difficult decisions (agree) | 31 (89%) |
| I have more insight in how I cope with conflicts (agree) | 35 (100%) |
| I am more aware of the patient's perspective (agree) | 24 (69%) |
| I am more aware of the family's perspective (agree) | 25 (71%) |
| I am more aware of the team's perspective (agree) | 29 (83%) |
| I am more aware of my internal barriers with regard to decisions at end-of-life (agree) | 31 (89%) |
| I am more aware of the external barriers with regard to decisions at end-of-life (agree) | 31 (89%) |
| **Effects of the coaching with regard to ethical decision-making: implementation in daily practice** | |
| The coaching sessions improved my skills in taking decisions with regard to end-of-life | 29 (83%) |
| I use what I have learned during the coaching sessions in my daily practice (agree) | 30 (86%) |
| I am able to implement insights from the coaching during the multidisciplinary meeting | 30 (86%) |
| I use the empowering leadership style during multidisciplinary meetings (agree) | 30 (86%) |
| The collaboration with senior doctors improved thanks to the coaching sessions (agree) | 21 (60%) |
| The collaboration with junior doctors improved thanks to the coaching sessions (agree) | 21 (60%) |
| The collaboration with the referring doctors improved thanks to the coaching sessions | 18 (51%) |
| The collaboration with the nurses improved thanks to the coaching sessions (agree) | 26 (74%) |
| **Effects of the coaching with regard to wellbeing** | |
| The coaching sessions were useful for myself (agree) | 34 (97%) |
| The coaching sessions increased my job satisfaction (agree) | 21 (60%) |

Table 3). Participating doctors expressed support for the continuation of the coaching. Most advocated for its integration throughout one's career, starting at the early beginning (Q17, Table 3). Yet, participants also discussed the problem that it is followed by people who are already prone to reflect (Q18, Table 3). The sting is how to reach the ones who are more reluctant, but who would benefit also. Mandating participation was questioned, as willingness to engage in introspection

**Table 3. Table of illustrative quotations about one-on-one coaching.**

| | Quotation | Discipline, Interview number |
|---|---|---|
| 1 | "And one of the things that was certainly discussed is how to deal with certain problematic situations, for example with nurses" | Doctor, FG3 |
| 2 | "Indeed as an individual, those coaching sessions were very valuable, just indeed especially that trusting relationship, sounding board, that you receive feedback, or if you are faced with a situation that you, if I say this or that, then you open up, get a different view." | Doctor, FG4 |
| 3 | "And often you are not aware of it. It is a kind of blind spot because you see it from the inside. It is only through the coach that you get an outside view and that you look at yourself from the outside, that you see your habits and that you can take a new approach.." | Doctor, FG3 |
| 4 | "There I could more easily reflect in a kind of safe way. When I talk to doctors, yes, you often hear different medical opinions there, just if that's retrospective, bon, that's basically it, you just don't doubt yourself. So I thought that was good. And yes, about your insecurities, yes… I thought that was good." | Doctor, FG3 |
| 5 | "After the sessions I have realized that it really took a while for myself, actually before that you really, I don't know if that is enough trust; but I think that is also the case for many other people in our position that you are not really used to make yourself vulnerable, that you would rather avoid that." | Doctor, FG1 |
| 6 | "It is like a safe heaven, a safe environment in which you can say your own opinion without being "punished" – although that's a big word – but… you can reflect in a safe manner" | Doctor, FG4 |
| 7 | "Because yes, we as doctors can sometimes be very insecure. That is the vulnerability that we often do not dare to show, but from that perspective I found the coaching conversations interesting. It allowed us to reflect, in a calm, relaxed way, not only about the case but also about how we deal problems, doubts and hesitations." | Doctor, FG8 |
| 8 | "And now I have learned to ask, if I even feel the slightest bit of oops, there is something exactly like that, or they have a different opinion, or they are quiet after a conversation, … do you agree? And that actually helps." | Doctor, FG4 |
| 9 | "Opening up the conversation with the patient. And asking more often: what effect has all this on you? How are you coping?" | Doctor, FG8 |
| 10 | "somewhere the coaching sessions have taught me to name things, not too offensively, but more constructively, or yes, tools to substantiate that more […] to name it from your own feeling, from your own concern, will have a positive affect" | Doctor, FG4 |
| 11 | "It has learned me daring to be assertive in a friendly way and to express my opinion and also set my boundaries." | Doctor, IV2 |
| 12 | "And then in concrete terms, yes, it has certainly improved awareness, the need for those DNR conversations and to overcome that threshold, er, perhaps also to move the priorities a little more in daily practice." | Doctor, FG1 |
| 13 | "Contrary to what may often be the climate, hey, not to get too involved in it [team dynamics], to now effectively say okay, play the ball faster if there is a signal like that somewhere, if you catch a signal like that, I have certainly gained from it." | Doctor, FG1 |
| 14 | "There is a tendency to quickly fall back into your habits." | Doctor, FG3 |
| 15 | It is very warm to see that it is not only about the family and the patient, but that our (nurses) opinion, I will put it this way, is also asked and that we are really involved in it." | Nurse, FG2 |
| 16 | "But they (doctors who participated in the coaching program) have grown in their communication and in their way of interacting with us, and especially in how they included us (nurses) in this, that is absolutely true." | Nurse, FG2 |
| 17 | "It wouldn't be a bad idea that such coaching sessions would we planned in the course of a career, or when the need is high, or even on a structural base." | Doctor, FG8 |
| 18 | "I think it is useful for everyone, but, I think it scares some people because you have to talk in private. And you expose yourself at a certain moment. And not everyone likes that." | Doctor, FG8 |
| 19 | "Or even a joint coaching trajectory maybe. Because, yes, they do not feel the changes. They pressed the alert, yet they do not directly feel what has changed." | Doctor, FG8 |
| 20 | "Not only coaching nurses individually, but actually coaching the dialogue together. Because I think by coaching a doctor individually, and by coaching the nurse individually, yes, okay, than you learn the skills, but in an ideal situation two people come together, and are coached on that spot." | Nurse, FG7 |

was mentioned as a prerequisite for a successful coaching trajectory. In thinking about the future of coaching the suggestion was raised both by doctors and nurses that nurses should also be involved in a sort of coaching trajectory, either separately or jointly (Q19-20, Table 3).

**The CODE alert.** *The use of the CODE alert:* Nurses testified that they used the alert in cases of a **feeling of powerlessness and not being heard by medical staff**. In some wards it was a tremendous relief to have a tool to report such situations; even so that the mere fact to have such a tool was already such a step forward that they didn't even expect doctors to do anything with it (Q1-2, Table 4). Pushing the alert was **never an impulsive act**. It was often accompanied by a round of deliberation with colleagues, even over the borders of work shifts or wards (Q3, Table 4). On some wards pushing the alert became a group event: they agreed to push the alert with several colleagues (Q4, Table 4). Many nurses articulated a certain caution in using the alert. Some out of fear it was not anonymous, or to be held accountable for it, some out of fear to disrupt a working relationship with their doctors. They did not want to finger point someone on the one hand, yet had a sense that what was going on didn't help the patient on the other hand. Pushing the alert took courage (Q5-6, Table 4).

*The perceived impact of the CODE alert on doctors and nurses:* On some wards nurses felt that the existence of the alert was a motivator for doctors to involve nurses in end-of-life decisions. The latter entailed that **doctors more often explained their motives** for a certain treatment decision, which nurses highly appreciated. Nurses testified that some **doctors more frequently inquired about nurses' view** and even when this involvement did not change the actual treatment choices, nurses found the care for that patient more enjoyable afterwards (Q7, Table 4). There were also wards in which nurses did not feel any effect of activating the CODE alert, which sometimes led to frustration and **increased feelings of powerlessness** (Q8, Table 4).

When doctors spoke of the CODE alert, they said they had a love hate relationship with the alert. They saw it as an accessible way for nurses to communicate on the one hand, yet they also regretted that the messenger hadn't found a direct way to communicate (Q9-10, Table 4).

Due to the CODE alert there was a **change in awareness and communication culture among nurses.** A kind of responsibility was awakened, as nurses sensed that they now had a tool to signal when they were too little involved (Q3,4,11,12, Table 4). Yet, there was also a concern articulated mostly by doctors, in that the CODE alert may never be narrowed to a frustration alert. It should always be accompanied by an argumentation and by a dialogue (Q13, Table 4). Furthermore it was argued that the alert must not only be used to debate a specific case but also as a sort of team thermometer (Q13, Table 4). The CODE study was a moment of sensitizing to install a culture of collaboration instead of one of pointing a finger. Nurses saw potential in a future CODE alert on the condition that direct feedback was visible to all parties. Participants saw an open dialogue as the ideal consequence of the alert. Yet, for some wards, this needed a non-involved third party to enable a safe talking environment (Q14, Table 4).

**Observation of the interdisciplinary team meetings.** The observation of interdisciplinary meetings was positively experienced because: there was a direct feedback loop after the observation; during the feedback also positive aspects were highlighted (Q15, Table 4); being real-life observation instead of second-hand talking about it (Q15, Table 4).

In general, the participants described the effects of the CODE study as opening up the visor. In some teams the CODE study opened up the dialogue between team members: they more easily debated cases in a non-threatening way (Q16, Table 4). There was even mentioned that the CODE study had ensued in a change of atmosphere on the ward (Q17, Table 4). Yet to attain this impact in a specific ward all the organization levels should click: the personal, team, and cultural level. It asks for seeing each other as equivalent partners with the same goal, namely the best possible care for the patients (Q18, Table 4).

## Results of the EDMCQ across roles

The (communication) gap between doctors and nurses popped up as one of the main results of the qualitative study and was confirmed by the sub-analysis of the EDMCQ [21], zooming in on the difference between doctors and nurses in the

**Table 4. Illustrative quotations about the CODE alert, the interdisciplinary meeting and CODE in general.**

| | Quotation | Discipline, Interview number |
|---|---|---|
| 1 | "I will communicate it directly to the doctors, but when that doesn't have any effect, I will not try further and then I will use the alert button. For me it is an option B. If it doesn't work in the direct way, I try the alert." | Nurse, IV1 |
| 2 | "The alert is when you as nurse have the feeling I am not heard." | Nurse, FG6 |
| 3 | "Sometimes that also happens if we also think – even if that is not our patient, but if that is a patient from, let's say, within our unit, and we also think that the care for that patient goes too far, then we sometimes also complete the code study, because then we know that if there are several of us who complete the study, you would hope that there would indeed be a little more response to that." | Nurse, FG5 |
| 4 | "The alert was usually pressed in consultation with colleagues, so that there were certainly always two colleagues and that it was looked into." | Nurse, FG5 |
| 5 | "I have already completed one (alert) myself, but there were actually several situations that would allow me to do so, but yes we are a bit reluctant. Because of the ignorant for sure. The guarantee of anonymity." | Nurse, IV1 |
| 6 | "You don't want to point a finger at someone, but if you feel that things are not going well for the patient: that's a real dilemma." | Nurse, IV1 |
| 7 | "They respect what we (nurses) say, what does not mean that they follow our opinion. But they effectively listen." | Nurse, FG6 |
| 8 | "In our team there was zero feedback after an alert. I heard the nurses complain that they pressed the alert and that they didn't have the impression that something was done." | Nurse, FG7 |
| 9 | "If I have to say one thing about that case in which the alert button was used, the thing that makes me feel a little offended is that it had to be done in such a way. That that person felt compelled to use the medium rather than discuss it directly with me." | Doctor, IV2 |
| 10 | "I think that is a bit a shame that after I explained something that the nurses did not come back to us that they didn't agree with it. And I actually think that's a shame that, while we are very open to it, that feedback was not returned. Instead they pushed the alert button." | Doctor, FG8 |
| 11 | "Nurses effectively reason more actively in the sense of: is this okay for the patient?, Can we ask the doctors to do something else? There is more dialogue and deliberation between nurses. And they support each other more." | Nurse, FG2 |
| 12 | "To get everyone involved in the story. Something that is very obvious or self-evident to us is not always so to someone else who is equally involved in the care of that patient. So I think that was the biggest added value for me in that study." | Doctor, IV2 |
| 13 | "I don't think it should be the way to work. That we push a button that signals to the other that there is something, I think we just need to communicate more openly and that it can teach us where our shortcomings lie." | Doctor, IV2 |
| 14 | "Actually, shortly after having pressed the alert button, you should know what happened with the report." | Nurse, FG7 |
| 15 | "It actually almost never happens that one is reinforced or that one gets the message that one had done something good. So yes, I'll say it, that's cool to hear that they, come on, if you then hear that yes, you're doing well or something... That's maybe the only thing I missed in the coaching sessions: the coach listens to all the things you want to tell, but there is no possibility to take her with you in real-life […] I maybe would have benefited even more when the coach would observe you and would then directly feedback what she had seen, because that are the things you often do not see yourself." | Doctor, FG1 |
| 16 | "I have the feeling that people are listening a bit more, yes. I no longer have that feeling that we are failing people. I do not know, there is a change of atmosphere or something, that there is more discussion." | Nurse, IV1 |
| 17 | "I just think the awareness that there is a need for better contact and if such sensitivities cannot be discussed, that there is a need for an alert, perhaps we should pay more attention to. I think that in itself that is the most important message, that we should make such things open for discussion." | Doctor, IV2 |
| 18 | "that there are more deep conversations, that we (nurses) see more their vision and that they (doctors) can see ours. […] Sit down, physically just sit down and just say, I saw that the code alert was pressed for that patient. | Nurse, IV1 |

evolution of the EDMCQ before and after the study period. As depicted in Fig 1, doctors estimated their "self-reflective and empowering leadership" skills higher than the nurses prior to the study period (P = 0.006) but no longer after the study period (P = 0.366). Nurses estimated "ethical awareness" and "open interdisciplinary reflection" higher (respectively, P = 0.002 and P = 0.049) and "the climate of not avoiding end-of-life decisions" and "active involvement of nurses at end-of-life" lower than the doctors (respectively P < 0.001 and P = 0.023) after the study period but not before the study period

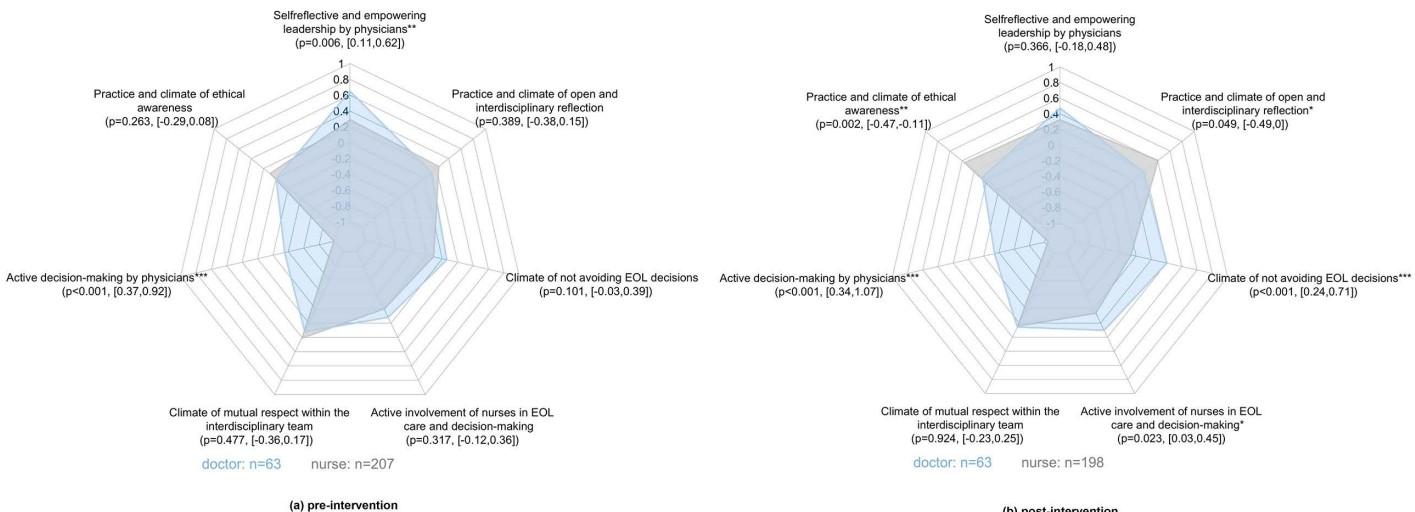

**Fig 1. Differences in EDMCQ scores between doctors and nurses before (a) and after (b) the intervention.** Sub-analysis of the validated ethical decision-making climate questionnaire (EDMCQ) across roles.

(all P-values >0.05). Nurses estimated the factor "active decision-making by doctors" much lower than doctors across both periods (both P<0.001).

## Discussion

The CODE study is the first intervention study that coached bedside doctors in self-reflective and empowering leadership as well as in managing group dynamics concerning ethical decision-making. This mixed-method evaluation study showed that the CODE intervention was successful in bringing awareness and opening visors among doctors and nurses concerning ethical decision-making. However, the CODE intervention did not close the communication gap between nurses and doctors.

### Process evaluation findings in relation to intervention effectiveness

This study showed that one-on-one coaching was an effective tool for installing (self-)reflection in doctors working in ten different acute hospital wards. That the majority of doctors testified of gained awareness and skills in decision-making is also reflected by more frequent treatment limitation decisions after the intervention in the quantitative analysis [17].

However, there is more work to be done to achieve true team-based decision-making; although this is widely recommended to achieve good clinical outcomes in complex patient situations [29–31]. The traditional role of the doctors as a medical expert ('the heroic lone healer' [3]) remains deeply ingrained in daily practice on hospital wards [3,32], as evidenced in this study. In order to set a safe and open climate to install team-based ethical decision-making, doctors will have to install vulnerability in their own practice when ethical matters are at stake in which there is not one science-based truth [33–35]. The need for a more safe and open climate to discuss ethical issues is also voiced by nurses who used the CODE alert to speak up their 'this-is-not-right' intuition [32,36–38]. There is a call for both nurses and doctors in this study to take responsibility in safe speaking-up: 'In case of continuation of the CODE alert it should be grounded in the responsibility of both parties: the (nurse) messenger who cannot activate the CODE alert without direct communication and argumentation and the (doctor) receiver who has to listen, explain and involve'. That the intervention did not succeed in involving and empowering the whole team in ethical decision-making is further supported by the EDMCQ sub-analysis between roles. The decrease in the factors "non-avoiding end-of-life decisions" and "active involvement of nurses at

end-of-life decision-making" in nurses after the study period suggests that the intervention did not fulfill the expectations of the nurses, and highlights the need to additionally coach the entire team.

To bridge the communication gaps between doctors and nurses, future iterations of the CODE intervention could consider implementing joint coaching sessions for doctors and nurses and/or interprofessional meetings focusing on real-life ethical dilemmas facilitating mutual understanding, respect and team-based decision-making [37,39–42].

### Strengths and limitations

The main strength is the complementarity of the findings with the main outcomes in the quantitative paper [17]. Methodological strengths of the process evaluation are (1) the high response rate for the satisfaction and EDMCQ survey and (2) the blinding for the qualitative study: the interviewer and main analyst were independent from the CODE intervention study, and the analyses for this study were finished before the main outcomes were known.

External validation of the results may pose a challenge. However, the high variability in ethical decision-making climates between the 10 participating wards found in the interview study as well as in the quantitative analyses [17] suggest that our findings may be extrapolated to other centers.

### Recommendations for practice and further research

In contrast to other investigators who focused on the effect of a consultant outside the team (communication facilitator, palliative or ethics consultant) [43–45], we preferred to perform an intervention in bedside clinicians to improve ethical decision-making. One-on-one coaching was successful in doctors, however longer coaching periods are recommended to solidify new attitudes, behaviors and insights into new habits [19,46]. Most doctors in this study were critical towards the sustainability of the gained skills and plead for continuation of the coaching as a normal practice, especially in younger doctors early in their careers [19,46].

Though, to implement change felt in the entire interprofessional team, future interventions should involve nurses and doctors together and provide workplace learning with closed-loop communication [31,46,47]. Additionally, expanding the intervention to multiple hospitals would enhance its generalizability and impact.

In future research, it is advisable to include the difference between doctors and nurses in EDMCQ scores rather than the overall EDMCQ alone.

## Conclusions

This mixed-method evaluation study showed that the CODE intervention was successful in bringing awareness and opening visors among doctors in ethical decision-making. The findings show nurses' increased awareness of their advocacy roles but also their frustration with the limited scope of their participation in ethical decision-making. Future interventions should prioritize closing the (communication) gap between doctors and nurses in order to attain ethical decision-making through the entire team to improve patient-centered care.

## Supporting information

**S1 Table. Characteristics of participants of the interview studies.**
(DOCX)

**S1 Appendix. COREQ (COnsolidated criteria for REporting Qualitative research) Checklist.**
(PDF)

**S1 File. English translation EC.**
(DOCX)

**S2 File. PLOSOne human subjects research checklist.**
(DOCX)

## Acknowledgments

We thank all of the participants who volunteered for this study as well as the heads (not included in the co-author list: Prof Sylvie Rottey, Medical Oncology; Prof Paul Boon, Neurology; Prof Steven Callens, Internal Medicine and Infectious Diseases; Prof Guy Brusselle Pneumology, doctors (Sofie Gevaert and Els Vandecasteele, Cardiology), chief nurses of the participating departments (Kristof Alluyn, Medical Oncology; Bart Sobrie, Internal Medicine; Isabelle Danel, Neurology; Els Carrijn, Geriatrics; Jo Vandenbossche, Medical Intensive Care Unit; Tania Helleputte, Gastro-intestinal and Liver Diseases; Fatima Snoussi, Pneumology; Katrien Schelfhout, Hematology; Gastroenterology and Pneumology; Annelies Masset, Cardiology and Internal Medicine; Jens Boelens, Cardiology Intensive Care; Christelle Lizy, Nephrology and Endocrinology; Delphine Lacante, Hematology) and care managers (Lieve De Geyter from the Man, Woman and Child Cluster; An Van Holsbeek from the Blood, Respiratory and Gastro-intestinal Cluster, Geert De Smet from the Metabolic and Cardiovascular Cluster and Hilde Goedertier from the Critical Care Cluster) for their commitment to this challenging study. We are also grateful to the nurses of the ICU trial cell (Anouska De Smeytere, Daisy Vermeiren, Jolien Van Hecke and Lesley Decoster) for their professional support. Last but not least, we would like to thank the following members of the Executive Board of the Ghent University Hospital to make this study possible; Prof Eric Mortier, CEO; Prof Frank Vermassen, Chief doctor; Prof. Rik Verhaeghe, Chief Nursing Officer; Prof Kristof Eeckloo, Responsible for the Strategic Policy, Quality Management, Information Management & Clinical Networks and Chantal Haeck, Responsible for the Company Supporting Service.

## Author contributions

**Conceptualization:** Ruth Piers, Katrijn Goethals, An Lievrouw, Karen Versluys, Frank Manesse, Stijn Vanheule, Dominique D. Benoit.

**Data curation:** Ruth Piers, Let Dillen, Katrijn Goethals, An Lievrouw, Karen Versluys, Dominique D. Benoit.

**Formal analysis:** Ruth Piers, Let Dillen.

**Funding acquisition:** Dominique D. Benoit.

**Investigation:** Ruth Piers, Let Dillen, Katrijn Goethals, An Lievrouw, Karen Versluys, Aglaja De Pauw, Celine Jacobs, Ine Moors, Fritz Offner, Anja Velghe, Nele Van Den Noortgate, Pieter Depuydt, Patrick Druwé, Dimitri Hemelsoet, Alfred Meurs, Jiska Malotaux, Wim Van Biesen, Francis Verbeke, Eric Derom, Dieter Stevens, Michel De Pauw, Fiona Tromp, Hans Van Vlierberghe, Karen Geboes, Frank Manesse, Dominique D. Benoit.

**Methodology:** Ruth Piers, Katrijn Goethals, An Lievrouw, Karen Versluys, Frank Manesse, Stijn Vanheule, Dominique D. Benoit.

**Project administration:** Ruth Piers, Aglaja De Pauw, Celine Jacobs, Ine Moors, Fritz Offner, Anja Velghe, Nele Van Den Noortgate, Pieter Depuydt, Patrick Druwé, Dimitri Hemelsoet, Alfred Meurs, Jiska Malotaux, Wim Van Biesen, Francis Verbeke, Eric Derom, Dieter Stevens, Michel De Pauw, Fiona Tromp, Hans Van Vlierberghe, Karen Geboes, Dominique D. Benoit.

**Supervision:** Ruth Piers, An Lievrouw, Frank Manesse, Stijn Vanheule, Dominique D. Benoit.

**Validation:** Ruth Piers, Let Dillen, Katrijn Goethals, An Lievrouw, Karen Versluys, Aglaja De Pauw, Celine Jacobs, Ine Moors, Fritz Offner, Anja Velghe, Nele Van Den Noortgate, Pieter Depuydt, Patrick Druwé, Dimitri Hemelsoet, Alfred Meurs, Jiska Malotaux, Wim Van Biesen, Francis Verbeke, Eric Derom, Dieter Stevens, Michel De Pauw, Fiona Tromp, Hans Van Vlierberghe, Karen Geboes, Frank Manesse, Dominique D. Benoit.

**Visualization:** Let Dillen, Dominique D. Benoit.

**Writing – original draft:** Ruth Piers, Let Dillen, Dominique D. Benoit.

**Writing – review & editing:** Ruth Piers, Let Dillen, Katrijn Goethals, An Lievrouw, Karen Versluys, Aglaja De Pauw, Celine Jacobs, Ine Moors, Fritz Offner, Anja Velghe, Nele Van Den Noortgate, Pieter Depuydt, Patrick Druwé, Dimitri Hemelsoet, Alfred Meurs, Jiska Malotaux, Wim Van Biesen, Francis Verbeke, Eric Derom, Dieter Stevens, Michel De Pauw, Fiona Tromp, Hans Van Vlierberghe, Karen Geboes, Frank Manesse, Stijn Vanheule, Dominique D. Benoit.

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
