## [Decision Letter · Decision Letter 0]

14 Nov 2025

Coaching Doctors to Improve Ethical Decision-Making in Adult Hospitalized Patients Potentially Receiving Excessive Treatment: Process evaluation study of the CODE intervention in Doctors and Nurses working in ten Acute Hospital wards

PONE-D-25-27745

Dear Dr. Piers,

We’re pleased to inform you that your manuscript has been judged scientifically suitable for publication and will be formally accepted for publication once it meets all outstanding technical requirements.

Kind regards,

Luca Valera

Academic Editor

PLOS ONE

Additional Editor Comments :

Congratulations! The referees expressed their satisfaction with your paper. Excellent work!

Reviewers' comments:

Reviewer's Responses to Questions

**Comments to the Author**

1. Is the manuscript technically sound, and do the data support the conclusions?

Reviewer #1: Yes

Reviewer #2: Yes

2. Has the statistical analysis been performed appropriately and rigorously?

Reviewer #1: Yes

Reviewer #2: Yes

3. Have the authors made all data underlying the findings in their manuscript fully available?

Reviewer #1: Yes

Reviewer #2: No

4. Is the manuscript presented in an intelligible fashion and written in standard English?

Reviewer #1: Yes

Reviewer #2: Yes

Reviewer #1: Dear Authors,

Your study has investigated a very important issue. Focus group research is very valuable in terms of providing difficult but very qualified data. It is very positive that nurses were included in the study along with physicians. Healthcare is a team effort and the awareness of each other's thoughts by employees can increase the quality of healthcare.

Thank you for your efforts.

Best regards

Reviewer #2: Outstanding article describing the qualitative research accompanying the CODE stepped-wedge cluster randomized controlled trial, which sheds light on the fact that the CODE intervention was successful and well-received by bedside doctors and nurses.

I have no comments whatsoever, other than to congratulate the investigators.

**Do you want your identity to be public for this peer review?** For information about this choice, including consent withdrawal, please see our Privacy Policy

Reviewer #1: No

Reviewer #2: No

---

## [Editor Report · Acceptance letter]

PONE-D-25-27745

PLOS ONE

Dear Dr. Piers,

I'm pleased to inform you that your manuscript has been deemed suitable for publication in PLOS ONE. Congratulations! Your manuscript is now being handed over to our production team.

Kind regards,

on behalf of

Dr. Luca Valera

Academic Editor

PLOS ONE